# Comparative genomic analysis of three geographical isolates from China reveals high genetic stability of Plutella xylostella granulovirus

**Man-Li Zhang, Ling-Qian Wang, Yong Qi, Yi Wu, Dong-Hui Zhou, Lu-Lin Li** *

Hubei Key Laboratory of Genetic Regulation and Integrative Biology, College of Life Sciences, Central China Normal University, Wuhan, China

* lilulin@mail.ccnu.edu.cn

**Data Availability Statement:** Sequence data of the PlxyGV isolates are available from Genbank (MN099284- MN099286).

## Abstract

In this study, the genomes of three Plutella xylostella granulovirus (PlxyGV) isolates, PlxyGV-W and PlxyGV-Wn from near Wuhan and PlxyGV-B from near Beijing, China were completely sequenced and comparatively analyzed to investigate genetic stability and diversity of PlxyGV. PlxyGV-W, PlxyGV-B and PlxyGV-Wn consist of 100,941bp, 100,972bp and 100,999bp in length with G + C compositions of 40.71–40.73%, respectively, and share nucleotide sequence identities of 99.5–99.8%. The three individual isolates contain 118 putative protein-encoding ORFs in common. PlxyGV-W, PlxyGV-B and PlxyGV-Wn have ten, nineteen and six nonsynonymous intra isolate nucleotide polymorphisms (NPs) in six, fourteen and five ORFs, respectively, including homologs of five DNA replication/late expression factors and two *per os* infectivity factors. There are seventeen nonsynonymous inter isolate NPs in seven ORFs between PlxyGV-W and PlxyGV-B, seventy three nonsynonymous NPs in forty seven ORFs between PlxyGV-W and PlxyGV-Wn, seventy seven nonsynonymous NPs in forty six ORFs between PlxyGV-B and PlxyGV-Wn. Alignment of the genome sequences of nine PlxyGV isolates sequenced up to date shows that the sequence homogeneity between the genomes are over 99.4%, with the exception of the genome of PlxyGV-SA from South Africa, which shares a sequence identity of 98.6–98.7% with the other ones. No events of gene gain/loss or translocations were observed. These results suggest that PlxyGV genome is fairly stable in nature. In addition, the transcription start sites and polyadenylation sites of thirteen PlxyGV-specific ORFs, conserved in all PlxyGV isolates, were identified by RACE analysis using mRNAs purified from larvae infected by PlxyGV-Wn, proving the PlxyGV-specific ORFs are all genuine genes.

## Introduction

Baculoviruses have long been explored as biological control agents of agricultural and forest pests attributing to their pathogenicity highly specific for insects, mainly Lepidoptera,

**Funding:** This study was funded by a grant from National Key R&D Program of China, 2017YFD0200400, (service.most.gov.cn) to LLL.

**Competing interests:** The authors have declared that no competing interests exist.

Hymenoptera and Diptera [1]. Genotypic variation in baculovirus populations have been widely detected between isolates from different geographical regions and within virus isolates by their genomic restriction endonuclease (REN) profiles in 1970s-1990s [2]. Differences in phenotypes were also revealed between different isolates and between genotypes derived from the single isolates by *in vitro* or *in vivo* techniques [2, 3]. More recently, hundreds of baculoviruses genomes have been completely sequenced, including the genomes of multiple different geographical isolates of some virus species. Nucleotide polymorphisms (NPs) have been documented between the different virus isolates and within the same isolate [3–8]. Comparative analysis of these genome sequences may make it possible to determine the genetic basis for phenotypic variations in populations of the same viral species. This may facilitate the improvement of baculovirus pesticides by mixing different virus genotypes. The codling moth, *Cydia pomonella*, was reported to be resistant to a Cydia pomonella granulovirus isolate CpGV-M. However, the resistance can be overcome by several other CpGV isolates. Whole-genome sequencing and phylogenetic analyses of these geographic CpGV variants revealed that the resistance is the consequence of a mutation in viral gene *pe38* [9, 10].

Plutella xylostella granulovirus (PlxyGV) belongs to the genus *Betabaculoviruses*. It is pathogenic for the diamondback moth, *Plutella xylostella*, a major destructive pest of cruciferous crops worldwide [11]. The virus has been isolated in several countries including Japan, China, India, Kenya, and South Africa [12–17]. In China, PlxyGV was first isolated, in Wuhan, and studied in 1970s [18]. Subsequently, it was isolated in other districts [19, 20]. Some strains of the diamondback moth have developed resistance to chemical pesticides and also become resistant to the bacterial insecticide *Bacillus thuringiensis* that have been used for its control [21, 22]. As an alternative, PlxyGV has been tested for the control of the pest [16, 19, 20, 23]. A registered PlxyGV biopesticide has been commercialized and used in large scale for the control of diamondback moth in China and Malaysia since 2008 [24–29]. Laboratory experiments have also been done to characterize PlxyGV morphology, histopathology, *in vitro* replication in cell culture, and molecular biology [17, 30–36].

PlxyGV has a single circular double-stranded DNA genome. The complete genome sequence of a PlxyGV isolate (PlxyGV-K1) from Japan was first reported in 2000 to consist of 100,999 bp and encode 120 putative protein-coding open reading frames (ORFs) [37]. Subsequently, the complete genome sequences of five additional isolates from mainland China (PlxyGV-C and PlxyGV-T), Taiwan (PlxyGV-K), Malaysia (PlxyGV-M), and South Africa (PlxyGV-SA) were published in 2016 [38, 39].

In this study, the genomes of three PlxyGV isolates named PlxyGV-W, PlxyGV-B and PlxyGV-Wn were completely sequenced. Intra isolate NPs and inter isolate NPs in the genomes were detected; And their insecticidal activity to the larvae of diamondback moth were also evaluated. In order to investigate genetic diversity and stability of PlxyGV, the genome sequences of these PlxyGV isolates and six previously reported PlxyGV genome sequences were comparatively analyzed, in nucleotide sequence variations, non-synonymous sequence polymorphisms, gene content and phylogeny.

## Materials and methods

### Virus and insects

The PlxyGV-W and PlxyGV-Wn were isolated near Wuhan in 1979 and 2018 from diseased *P. xylostella* larvae in cabbage fields, respectively [18]. PlxyGV-B is from a commercialized biopesticide, that was originally isolated in Beijing in 1980s. The isolates were propagated by feeding an artificial diet contaminated with the virus occlusion bodies (OBs) to third instar laboratory reared Diamondback moth larvae.

Purification of OBs and extraction of viral DNA were carried out as described by Hashimoto et al. [40], with modifications. Approximately100 infected larvae were homogenized with a blender. The worm-tissue fragments in the homogenate were removed by differential centrifugation at 750 RPM for 15min and 8,500 RPM for 25min, and repeated twice. The pellet was suspended in 3–4 ml of ddH$_2$O. The suspension was layered onto a 30, 40, 50, and 60% (wt/vol) discontinuous sucrose gradient and centrifuged at 4,000 rpm for 1h. The OB fraction was collected and washed twice by suspension in H$_2$O and centrifugation. The OBs were dissolved in equal volume of alkaline solution (100 mmol/L NaCl, 100 mmol/L Na$_2$CO$_3$, 5 mmol/L EDTA) and incubated at room temperature with stirring, then mixed with equal volume of protein digestive solution [1 mmol/L EDTA, 1% SDS (w/v), 10 mmol/L Tris-HCl (PH 7.4), 0.5 mmol/L NaCl, 0.2 mg/L protease K], incubated in a bath at 58˚C overnight. Viral DNA was extracted twice with an equal volume of phenol-chloroform and precipitated by mixing with ethanol centrifugation, then dissolved in TE buffer and stored at 4˚C.

## Genome sequencing

The genomes of the three PlxyGV isolates were sequenced by using Illumina Hisen X ten system. Sequence assembly were done by using SOAP denovo (Version 2.04) software (BGI), and using the first published genome sequence of the PlxyGV-K1 isolated from Japan (NC_002593) as a reference. PCR was performed to synthesize DNA fragments bridging the gaps between contigs by using the genomic DNAs of individual PlxyGV isolates as templates. PCR products were sequenced from both ends. The sequences were assembled with the initial contigs into a single, circular contig. Sequences were analyzed with Lasergene programs (DNASTAR). Homology searches were carried out with GenBank/EMBL, SWISSPROT and PIR databases by using the BLAST algorithm. Multiple sequence alignments were performed by using CLUSTAL W. The PlxyGV genome sequence accession numbers are MN099284 for PlxyGV-W, MN099285 for PlxyGV-B and MN099286 for PlxyGV-Wn.

## RNA purification and RACE analysis of PlxyGV-specific genes

*P. xylostella* larvae in third instar were infected with PlxyGV-Wn by feeding with viral OBs-contaminated diet and collected at 12 h, 24 h, 48 h, 72 h and 96 h post infection. 25 infected larvae (five larva from each time point) were immersed and homogenized in 1,000 μl of Trizol and incubated on ice for 10 min, then centrifuged at 11,400 rpm and 4˚C for 10 min. The supernatant was mixed with 200 ml of chloroform with shaking for 15 s, and incubated on ice for 15 min, then centrifuged at 11,400 rpm and 4˚C for 15 min. 400 μl of the upper phase was taken and mixed with 500 μl of isopropyl alcohol, incubated at room temperature, then centrifuged at 11,400 rpm for 10 min. The pellete was rinsed with 200 μl of 75% ethanol in DEPC water by centrifugation at 11,400 rpm for 5 min, air dried, then dissolved in x μl of DEPC water. 8 μl of RNA sample was mixed with 1 μl of DNA digestion buffer and 1 μl of DNase I, and incubated at 37˚C for 30 min. Then 2 μl of 50 mM EDTA was added to inactivated DNase I by incubation at 65˚C for 10 min.

First-strand cDNAs synthesis and RACE were done by using SMARTer® RACE 5′/3′Kit following manufacturer's instruction. For first-strand cDNAs synthesis, mixed 4.0 μl of 5x first-strand buffer, 0.5 μl DTT (100mM) and 1.0 μl dNTPs (20 mM) to make reaction buffer mix, at first. Send, mixed 1.0 –10 μl of RNA, 1.0μl 5'-CDS primer A (or 3'-CDS primer A) and 0 –10μl DEPC water, incubated the mixtures at 72˚C for 3 min, cooled down at 42˚Cfor 2 min, then spun at 14,000g for 10s, to make denatured RNAs. To the RNA sample for 5'RACE cDNA synthesis, add 1 μl of SMART II A Oligonucleotide. Then, mixed 5.5 μl of 5'RACE or 3'RACE cDNA synthesis reaction buffer mix, 0.5 μl RNase inhibitor (40 U/μl)

and 2.0 μl SMARTScribe Reverse Transcripase (100 U), to make 5'RACE or 3'RACE cDNA synthesis master mix. Finally, combined 8 μl of the 5'RACE or 3'RACE cDNA synthesis master mix with the denature RNA sample, incubated at 42˚C for 90 min, then heated at 72˚C for 10 min. Diluted the sample by adding 10 μl (if started with $\leq$ 200 ng of total RNA) or 100 μl (if started with$\geqq$200 ng of total RNA) of Tricine-EDTA buffer to make 5' RACE-ready cDNA and 3'-RACE-ready cDNA. To perform RACE, mixed 25 μl of 2x seqAmp buffer, 2.5 μl of 5'(3') RACE-ready cDNA, 5 μl of 10x universal primer A mixture, 1 μl of 5'(3') RACE primer, 1 μl of SeqAmp DNA polymerase and 15.5 μl of PCR-grade water, and ran PCR as below: first step: 94˚C,4 min. Second: 94˚C, 30s; 68˚C, 30s; 72˚C, 3min; 25 cycles. Third: 72˚C, 7 min.

### Bioassays

Bioassays on the infectivity of PlxyGV isolates were performed as previously described [41]. To determine the median lethal concentration ($LC_{50}$), virus suspensions in concentrations of 0, $1\times10^7$, $2\times10^7$, $5\times10^7$, and $1\times10^8$ OBs/ml were prepared respectively, by suspending the virus OBs in 4% sucrose in double-distilled water containing 0.05% food blue dye. Quantification of PlxyGV was performed by using a qPCR method. The virus suspensions were used to feed newly molted third-instar *P. xylotella* larvae that had been starved for twelve hours. The larvae that had swallowed the virus suspension were picked and transferred into the wells of twelve-well plates and feed with fresh artificial diet for the duration of the bioassay. Mortality was recorded daily after infection until larvae died or pupated. Forty eight larvae per concentration were used in the infection experiments and the experiments were repeated in triplicate. The $LC_{50}$ values were determined by the probit analysis calculated compared with a relative median potency method. To determine median lethal time ($ST_{50}$), newly molted third-instar *P. xylotella* larvae were oral infected in the same way as above, using virus suspensions of $1\times10^9$ OBs/ml. Mortality was recorded every 6 h after infection until all larvae died or pupated. The $ST_{50}$ values were calculated with the Kaplan–Meier estimator and compared by the log-rank test.

## Results

### Genome sequences of PlxyGV-W, PlxyGV-B and PlxyGV-Wn

The genomes of PlxyGV-W, PlxyGV-B and PlxyGV-Wn are 100,941 bp, 100,972 bp and 100,999 bp in length with a G + C compositions of 40.73%, 40.71% and 40.71%, respectively. A complete sequence alignment showed a sequence identity of 99.8% between PlxyGV-W and PlxyGV-B, 99.6% between PlxyGV-W and PlxyGV-Wn, and 99.5% between PlxyGV-B and PlxyGV-Wn, respectively. The gene contents, genome organization and variations of the three isolates are demonstrated by Fig 1. It is shown that all the three PlxyGV isolates contains 118 ORFs in common, being 150 bp or longer, starting with an ATG and having minimal overlap with adjacent ORFs or homologous repeat regions (*hr*s), respectively. All the homologous ORFs and *hr*s are completely collinear in organization in the three isolates.

Intra-isolate NPs are detected in all the three virus isolates. In PlxyGV-W genome, thirty two single nucleotide polymorphisms (SNPs) and four NPs involving two or more nucleotide alterations are identified. Although the majority of the NPs locate in ORFs, only eight SNPs encode amino acid alterations, occurring in ORF26 (*f*), ORF32 (*lef2*), ORF61 (*dbp*), ORF99 (*lef9*), ORF104 (*fgf*), ORF109 (*lef8*) and ORF113, respectively, and there is a NP with a insertion/deletion (InDel) of twelve nucleotides after AA68 and a deletion of a single nucleotide causing frame shift after AA101, in ORF73 (*ac91*) (Table 1).

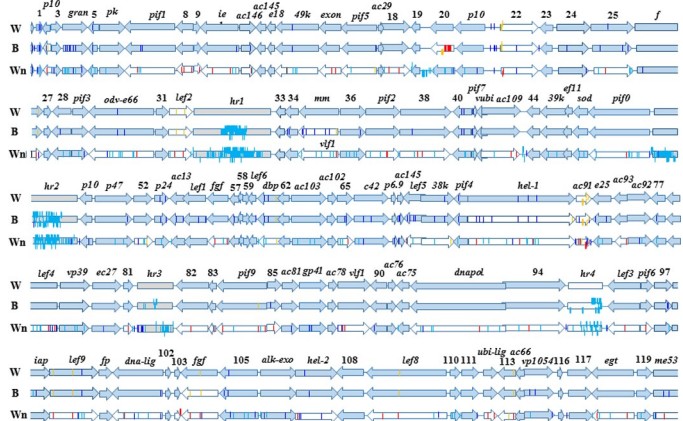

**Fig 1. Comparison of genome structure between PlxyGV-W, PlxyGV-B, and PlxyGV-Wn.** The figure depicts a schematic representation of the PlxyGV genomes with map positions of the 118 ORFs represented by arrows indicating transcriptional direction and relative size. Numbers above arrows represent the number of each ORF. Blue and orange vertical lines represent the locations of synonymous and nonsynonymous nucleotide polymorphisms respectively. Light blue and red vertical lines represent the locations of synonymous and nonsynonymous point mutations in the PlxyGV-B and PlxyGV, compared to PlxyGV-W. Raised lines represent insertions and the lowered lines represent deletions. Turning lines represent frameshift mutations. *hr* sequences and their positions on the genome are indicated by light grey boxes. Empty arrows represent ORFs with nonsynonymous mutations relative to PlxyGV-W.

In the PlxyGV-B genome, there are 105 SNPs and six NPs involving multiple nucleotide alterations. The majority of the NPs identified in PlxyGV-W are also found in PlxyGV-B. Nineteen NPs causing amino acid changes in thirteen ORFs of PlxyGV-B genomes including ORF2 (*p10*), ORF20, *f*, *lef2*, ORF35 (*mmp*), *dbp*, ORF68 (*ac145*), ORF72 (*hel-1*), ORF73 (*ac91*), ORF84 (*pif8*), *lef9*, *fgf* and *lef8* and ORF112 (*ac53*). *lef2*, *lef9* and *fgf* contain two, and ORF73 contains three nonsynonymous NPs (Table 1). Similar to PlxyGV-W, there is an identical NP with an InDel of twelve nucleotides and an InDel of a single nucleotide causing frame shift in ORF73. The difference is that the sequencing reads missing the twelve nucleotides are more than the ones with the twelve nucleotides, in PlxyGV-W ORF73. In contrast, the sequencing reads missing the twelve nucleotides are much less than the ones with the twelve nucleotides, in PlxyGV-B ORF73. In PlxyGV-B ORF20, there is an NP involving an InDel of eighteen nucleotides. PlxyGV-Wn genome contains sixty nine SNP sites. Only six SNPs in five ORFs induce amino acid alterations, including ORF16 (*pif5*), ORF52 (*ac38*), ORF70 (*38k*), ORF73 (2 SNPs) and ORF113 (Table 1). Most SNPs in PlxyGV-Wn genome are different from the ones in PlxyGV-W and PlxyGV-B genomes. Majority of the SNPs in all the three PlxyGV isolates are nucleotide transitions.

## Comparison of the genomes of PlxyGV-W and PlxyGV-B

PlxyGV-B genome is 31 bp longer than PlxyGV-W genome. The difference is mainly in the *hr* regions. The *hr1*, *hr2*, *hr3*, and *hr4* of PlxyGV-B are 40 bp, 2 bp, 3 bp and 15 bp longer than those of PlxyGV-W, respectively. The sizes of all putative protein-coding ORFs of PlxyGV-B are same as the ones of PlxyGV-W except ORF20 and ORF73 (Table 2). Relative to PlxyGV-W ORF20, PlxyGV-B ORF20 has six amino acids deleted after AA119. And there are twelve additional amino acid variations between these two ORF20 homologs. As mentioned above, both PlxyGV-W and PlxyGV-B ORF73 homologs have two NPs at AA68 and AA101 (PlxyGV-W)/94(PlxyGV-B) sites. In PlxyGV-W ORF73, most sequencing reads have the extra twelve nucleotides encoding "TPPP" after AA68, and a small part of sequencing reads miss the twelve

**Table 1. Intra-isolate nucleotide polymorphisms in PlxyGV-W, PlxyGV-B and PlxyGV-Wn.**

| ORF | Name/Dir | PlxyGV-W | | PlxyGV-B | | PlxyGV-Wn | |
|---|---|---|---|---|---|---|---|
| | | nt | aa | nt | aa | nt | aa |
| | | | | C152CG | | | |
| 1 | → | | | | | A225G | |
| | | G289A | | G289A | | | |
| | | G309A | | G309A | | | |
| 2 | 10/← | A391G | | A391G | | | |
| | | | | C486T | | | |
| | | | | C511A | M33I | | |
| 3 | → | | | | | C681T | |
| 4 | Gran/→ | | | G1031C | | | |
| | | | | A1064G | | | |
| | | | | T1154G | | | |
| | | | | A1175G | | | |
| | | | | C1274T | | | |
| | | | | | | T1280C | |
| | | | | C1391T | | | |
| | | | | | | A1520G | |
| 5 | ← | G1865T | | G1865T | | | |
| 7 | Pif1/← | | | T3443C | | | |
| | | | | A3458G | | | |
| 10 | ie1/← | | | T6368C | | | |
| 11 | ac146/→ | | | T6807C | | | |
| 14 | 49k/← | G8105A | | G8105A | | | |
| 16 | pif5/← | | | | | T10608G | K32T |
| 20 | ← | | | G12756GAGAAGGGCATCTTCGAGG | P89PRRCPSP | | |
| | | | | | | T13017A | |
| | | T14324G | | G14306T | | | |
| | | T14495A | | T14477A | | | |
| | | C14505T | | T14487C | | C14471T | |
| | | | | T14530A | | | |
| | | G14662GT | | G14644GT | | | |
| 22 | → | | | G15085A | | G15070A | |
| | | | | G15839A | | | |
| 24 | → | | | T16717C | | | |
| | | | | | | T17377C | |
| 25 | → | | | C17775T | | | |
| | | | | | | T18084C | |
| | | | | A18519G | | | |
| | | | | C18690G | | | |
| 26 | f/→ | | | | | A19111C | |
| | | T20360G | L494R | T20342G | L494R | | |
| | | | | | | G20349T | |
| 28 | ← | | | G21056A | | | |
| 29 | pif3/→ | | | | | A21832G | |
| 30 | odv-e66/← | | | | | G22527A | |
| | | C22992T | | C22974T | | | |
| | | | | A23208G | | | |

(*Continued*)

**Table 1.** (*Continued*)

| ORF | Name/Dir | PlxyGV-W | | PlxyGV-B | | PlxyGV-Wn | |
|---|---|---|---|---|---|---|---|
| | | nt | aa | nt | aa | nt | aa |
| 32 | *lef2/*→ | G24631A | D69N | G24613A | D69N | | |
| | | | | C24927A | N173K | | |
| | | | | | | C27728T | |
| 34 | ← | | | G28191A | | | |
| | | | | G28270A | | | |
| 35 | *Mmp/*← | | | T28615C | | | |
| | | | | A28659G | | | |
| | | | | T28702C | | | |
| | | | | A29080G | | | |
| | | | | C29398T | | | |
| | | | | A29500C | | | |
| | | | | T29721C | M10V | | |
| 36 | *p13/*→ | | | C30478T | | | |
| 39 | → | | | A32122G | | | |
| 40 | *ac106/107/*← | C33552T | | C33574T | | | |
| | | G33696A | | G33718A | | | |
| 41 | *pif7/*← | C33910T | | C33932T | | | |
| | | | | | | T35671G | |
| | | | | | | T39530G | |
| *hr2* | | | | | | G39699T | |
| | | | | CT40366C | | | |
| | | | | | | C41711T | |
| | | | | | | C41929G | |
| | | | | | | A41932G | |
| 50 | *p10/*← | | | | | C42272T | |
| 51 | *p47/*→ | | | T43504C | | | |
| 52 | *ac38/*→ | | | | | T43829C | |
| | | | | | | C44078T | R139K |
| 53 | *p24/*→ | A44488T | | T44512A | | | |
| | | A44692T | | A44716T | | | |
| 54 | *ac13/*← | | | G45080A | | | |
| | | | | G45213T | | | |
| 55 | *lef1/*← | | | G45666A | | | |
| | | | | A45729T | | | |
| 57 | ← | | | | | C46734T | |
| | | | | G47492A | | | |
| 61 | *Dbp/*← | A47648G | | A47672G | | | |
| | | T47735C | | T47759C | | | |
| | | T48028C | I82V | C48052T | V82I | | |
| 65 | → | | | G50238C | | | |
| | | | | A50279C | | | |
| 66 | *ac101/*→ | | | A50703G | | A50729G | |
| | | | | C51015A | | | |
| 67 | *p6.9/*→ | | | | | G51747A | |
| 68 | *ac145/*→ | | | A51839ATTG | | | |
| | | | | G51857A | | | |

(*Continued*)

**Table 1.** (*Continued*)

| ORF | Name/Dir | PlxyGV-W | | PlxyGV-B | | PlxyGV-Wn | |
|---|---|---|---|---|---|---|---|
| | | nt | aa | nt | aa | nt | aa |
| 69 | *lef5/←* | | | T52117C | | | |
| | | | | G52152T | | | |
| | | | | A52296G | | | |
| | | | | A52371G | | | |
| 70 | *38k/→* | | | T52976C | | | |
| | | C53271T | | C53295T | | | |
| | | | | | | A53545G | K321E |
| 71 | *pif4/←* | A53664G | | A53688G | | | |
| 72 | *hel1/→* | | | G54288A | | | |
| | | | | C54366T | | | |
| | | | | A54513G | | | |
| | | | | A54675G | | | |
| | | | | | | G55573A | |
| | | C55711T | | C55735T | | | |
| | | A56019G | | A56043G | | | |
| | | C56208T | | C56232T | | | |
| | | | | G57350A | K1110R | | |
| 73 | *ac91/→* | | | A57477C | | | |
| | | | | T57488C | T28I | | |
| | | | | | | C57553T | |
| | | | | | | A57592T | L54F |
| | | TCCACCGCCGACC57552-T | TPPPT68-72T | T57575TCCACCGCCGACC | T68TPPPT | T57602C | T69P |
| | | A57684AC | 101fram shift | AC57697A | 101fram shift | | |
| | | | | | | G57755A | |
| | | | | | | A57756C | |
| | | | | | | A57758T | |
| | | | | | | A57761T | |
| | | | | G57855A | | G57869A | |
| 74 | *odv-e25/←* | | | A58307T | | | |
| 76 | *ac92/→* | | | | | T59321C | |
| 77 | *←* | | | T59853C | | | |
| 80 | *odv-ec27/→* | | | | | A62986G | |
| | | | | | | G63984A | |
| *hr3* | | | | | | G64044C | |
| | | | | | | G64045T | |
| | | | | | | G64052A | |
| | | | | | | C64182T | |
| | | | | | | C64183T | |
| | | | | | | G64194A | |
| 84 | *pif8/←* | | | T67695C | V109T | | |
| 85 | *tlp20/→* | | | G68166A | | | |
| 87 | *gp41/→* | | | | | A69375T | |
| | | | | | | T69454G | |
| 89 | *vlf1/→* | | | | | C70312T | |
| 90 | *←* | | | | | C71254T | |
| 93 | *Dnapol/←* | | | | | G74855A | |

(*Continued*)

**Table 1.** (Continued)

| ORF | Name/Dir | PlxyGV-W | | PlxyGV-B | | PlxyGV-Wn | |
|---|---|---|---|---|---|---|---|
| | | nt | aa | nt | aa | nt | aa |
| | | | | | | G77305A | |
| hr4 | | TTGTTAAAATAAATAAACATTC77980T | | | | | |
| | | | | | | T78637A | |
| 97 | → | | | A80551G | | | |
| | | | | C80554A | | | |
| 98 | Iap/→ | | | A80957G | | | |
| 99 | lef9/→ | G81575A | R50Q | G81606A | Q50R | | |
| | | T82198G | S258A | T82229G | A258S | | |
| | | | | | | T82326C | |
| | | G82434A | | G82465A | | | |
| 101 | dna-lig/← | | | | | A84127G | |
| | | | | T84914C | | T84941C | |
| 104 | Fgf/← | | | C85724T | I318V | | |
| | | A86038G | F203L | A86069G | F203L | | |
| 105 | ← | | | G86805A | | | |
| | | G86891A | | G86922A | | | |
| | | | | | | A86996G | |
| | | | | | | C87801T | |
| 106 | alk-exo/→ | | | | | A88162G | |
| | | | | | | C88804A | |
| 107 | hel2/→ | | | | | C89169T | |
| | | | | | | G89201A | |
| | | | | | | G89858A | |
| | | | | G89527A | | | |
| | | G90072A | | | | | |
| 109 | lef8/← | T92293C | F655L | T92324C | F655L | | |
| | | | | | | A93448G | |
| 112 | ac53/→ | | | | | A95747G | F79L |
| | | A95854G | V24A | A95885G | V24A | | |
| 115 | vp1054/→ | | | C96394T | | | |
| | | | | C96644T | | | |
| | | | | | | T97116C | |
| | | | | | | A97526G | |
| 120 | me53/→ | | | | | C100273T | |
| | | | | A100611G | | | |
| | | | | A100788G | | | |
| | | | | T34C | | C34T | |

*Nucleotide polymorphisms (nt) and amino acid changes (aa) encoded are presented. The numbers between letters indicate the positions in the genomes or in ORFs.

nucleotides. In contrast, most sequencing reads miss the twelve nucleotides and small parts of sequencing reads have the ones in PlxyGV-B ORF73. At the AA101/97 site, most sequencing reads contains an "A", small part of sequencing reads contains "AC" that makes a frame shift. In contrast, most sequencing reads contains "AC" and small part of reads contain "A", in PlxyGV-B ORF73. There are seven additional nonsynonymous variations in six ORFs between PlxyGV-B and PlxyGV-W (Fig 1 and Table 2).

**Table 2. Nonsynonymous mutations in PlxyGV-W, PlxyGV-B and PlxyGV-Wn.**

| ORF | Name | Size (aa) | PlxyGV-W | PlxyGV-B | PlxyGV-Wn |
|---|---|---|---|---|---|
| 2 | P10 | 83 | 49S | 49S | 49N[@] |
| 7 | PIF1 | 536 | 385R | 385R | 385K |
| 8 | | 175 | 18V, 110N, 137A | 18V, 110N, 137A | 18I, 110T, 137T |
| 9 | | 80 | 62I | 62I | 62M |
| 10 | IE1 | 393 | 129D, 320G | 129D, 320G | 129G, 320E |
| 11 | AC146 | 180 | 116V | 116V | 116A |
| 15 | EXON0 | 216 | 45I, 120L | 45I, 120L | 45M, 120S |
| 16 | PIF5 | 351 | 32T | 32T | 32K |
| 18 | | 313 | 102T, 185V | 102T, 185V | 102P, 185A |
| 20 | | 229/223/229 | 52C, 57W, 66S, 69L, 79LI, 88RRRR, 94A, 114RCPSPR | 52N, 57E, 66C, 69A, 79SL, 88PPSC, 94P, 114∧6 | 52C, 57W, 66S, 69L, 79LI, 88RRRR, 94A, 114RCPSPR |
| 25 | | 431 | 134E | 134E | 134K |
| 26 | F | 553 | 494L | 494L | 494R |
| 30 | ODV-E66 | 682 | 205F | 205F | 205Y |
| 32 | LEF2 | 270 | 69D, 173N | 69D, 173N/K | 69N, 173K |
| 35 | MMP | 402 | 10M | 10M | 10V |
| 37 | PIF2 | 368 | 199F, 202S | 199F, 202S | 199L, 202G |
| 38 | | 528 | 181K, 206Q | 181K, 206Q | 181E, 206K |
| 43 | AC109 | 414 | 122V, 381S | 122V, 381S | 122A, 381T |
| 49 | PIF0 | 627 | 286A | 286A | 286T |
| 56 | FGF | 221 | 6S, 149D | 6S, 149D | 6Y, 149E |
| 61 | DBP | 263 | 82I | 82V | 82V |
| 65 | | 152 | 115E | 115E | 115K |
| 69 | LEF5 | 247 | 212R | 212R | 212K |
| 70 | 38K | 340 | 321E | 321E | 321K |
| 72 | HEL1 | 1124 | 97I, 531N, 665N, 1110R | 97I, 531N, 665N, 1110K/R | 97N, 531S, 665D, 1110R |
| 73 | AC91 | 153/159/155 | 28I, 54F, 69TPPP/∧, 103-153fs/n | 28T/I, 54F, 69∧4/TPPP, 103∨PPPPPPPP | 28T, 54L/F, 69T/PPPP, 103∧8 |
| 75 | AC93 | 156 | 24D | 24D | 24E |
| 78 | LEF4 | 432 | 32D, 91I | 32D, 91I | 32E, 91T |
| 81 | | 107 | 43W | 43W | 43G |
| 82 | | 340 | 232Y | 232Y | 232C |
| 83 | | 69 | 32R | 32R | 32K |
| 84 | PIF8 | 533 | 66H, 109I, 481S | 66H, 109V 481S | 66Y, 109V, 481A |
| 85 | TLP20 | 139 | 71C | 71C | 71S |
| 89 | VLF1 | 346 | 75I,89N, | 75I,89N, | 75V, 89K |
| 93 | DNAPOL | 978 | 345K | 345K | 345R |
| 94 | AC66 | 651 | 291A | 291A | 291S |
| 95 | LEF3 | 337 | 179R,180I | 179R,180I | 179K, 180M, |
| 96 | PIF6 | 128 | 123R, | 123R, | 123H |
| 99 | LEF9 | 494 | 50R, 258S | 50Q, 258A | 50Q, 258A |
| 101 | DNA-LIG | 523 | 383V | 383V | 383I |
| 103 | | 63/63/66 | | | 63∨LKK |
| 104 | FGF | 396 | 203F,318V | 203F, 318I | 203L, 318I |
| 107 | HEL2 | 436 | 12S | 12S | 12G |
| 109 | LEF8 | 838 | 479I, 655F | 479I, 655F | 479M, 655L |
| 112 | UBI-LIG | 137 | 61I,117T | 61I,117T | 61M, 117I |

*(Continued)*

**Table 2.** (Continued)

| ORF | Name | Size (aa) | PlxyGV-W | PlxyGV-B | PlxyGV-Wn |
|-----|------|-----------|----------|----------|-----------|
| 113 |      | 198       | 24V,79L  | 24V,79L  | 24A, 79F  |
| 118 | EGT  | 429       | 69Q, 394N | 69Q, 394N | 69K, 394D |
| 120 | ME53 | 260       | 134R     | 134R     | 134G      |

@ The numbers in front the letters indicate the positions of mutations; ∨ indicates insertion; ∧ indicates deletion; the numbers following ∧ indicate number of codons missing; fs, frameshift; Single isolate specific variations are printed in grey color.

## Comparison of PlxyGV-Wn genome with the genomes of PlxyGV-W and PlxyGV-B

PlxyGV-Wn genome is 58 bp and 27 bp longer than the genomes of PlxyGV-W and PlxyGV-B, respectively. Similarly, the differences are also mainly due to the differences in the length of the *hrs*. There are totally 486 nucleotide substitutions between the genomes of PlxyGV-Wn and PlxyGV-W, including seventy three nonsynonymous point mutations in forty seven ORFs (Table 1 and Fig 1). Twelve positions with nonsynonymous point mutation have polymorphisms in either both or one virus isolate. In addition, there are 262 nucleotide InDels. Relative to PlxyGV-Wn ORF73, PlxyGV-W ORF73 has an insertion of 23 nt after AA101, resulting in a frame shift at C-terminal. However, PlxyGV-W ORF73 has a polymorphism at this part as described above. A single nucleotide substitution at the 3'-end of ORF103 has the stop codon in PlxyGV-W changed to leucine codon and adds additional two codons to the C-terminal in PlxyGV-Wn. The sizes of the other ORFs are identical between PlxyGV-Wn and PlxyGV-W.

There are 469 nucleotide substitutions between the genomes of PlxyGV-Wn and PlxyGV-B. Relative to PlxyGV-B, PlxyGV-Wn genome has 144 nt insertions and 117 nt deletions. These nucleotide mutations encode seventy seven nonsynonymous changes in forty six ORFs. Majority of nonsynonymous variations between PlxyGV-Wn and PlxyGV-B are same as the ones between PlxyGV-Wn and PlxyGV-W with a few exceptions (Table 2 and Fig 1). Relative to PlxyGV-B, PlxyGV-Wn ORF73 has four codons inserted after AA68 and a cluster of eight proline codons deleted after AA95. Unlike PlxyGV-B ORF20 that contains five nonsynonymous mutations relative to PlxyGV-Wn ORF20, PlxyGV-W ORF20 encodes the same amino acid sequences as PlxyGV-Wn ORF20. ORF61 and ORF99 contain one and two nonsynonymous variations between PlxyGV-Wn and PlxyGV-W whereas there is no difference in these two ORFs between PlxyGV-Wn and PlxyGV-B.

## PlxyGV-W and PlxyGV-B demonstrate higher insecticidal activity than PlxyGV-Wn for *P. xylotella* larvae

The infectivity of PlxyGV-W, PlxyGV-B and PlxyGV-Wn were tested for newly molted third-instar *P. xylostella* larvae by feeding the larvae with viral OBs and determining $LC_{50}$ and $ST_{50}$ in bioassays. As shown in Table 3, the $LC_{50}$ of PlxyGV-Wn is about two times of the ones of the other two virus isolates while there is no significant difference between PlxyGV-W and PlxyGV-B. No significant difference is detected in $ST_{50}$ between all the three isolates at a concentration of $1 \times 10^9$ OBs/ml (Table 4).

## Comparison of the genome sequences of nine PlxyGV isolates

To investigate diversity of PlxyGV isolates from different area, the genome sequences of PlxyGV-W, PlxyGV-B, PlxyGV-Wn are compared with six additional complete PlxyGV

**Table 3. Concentration-mortality of PlxyGV-W, PlxyGV-B and PlxyGV-Wn for *Plutella xylostella* larve.**

| | Virus | LC$_{50}$ (95%CI) (×10$^7$OB/mL) | Relative median potency (95%CI) | |
| --- | --- | --- | --- | --- |
| | | | PlxyGV-W | PlxyGV-B |
| Test1 | PlxyGV-W | 1.695 (1.171–2.335) | - | - |
| | PlxyGV-B | 1.570 (1.080–2.162) | 0. 926 (0.583–1.457) | - |
| | PlxyGV-Wn | 3.131 (2.284–4.291) | 1.847 (1. 183–3.085) | 1.995 (1. 275–3.372) |
| Test2 | PlxyGV-W | 2.300 (1.318–4.155) | - | - |
| | PlxyGV-B | 3.375 (1.352–4.199) | 1.033 (0.472–2.374) | - |
| | PlxyGV-Wn | 4.516 (3.002–7.292) | 1.964 (0.979–5.903) | 1.901 (0.964–5.350) |
| Test3 | PlxyGV-W | 2.282 (1.393–3.840) | - | - |
| | PlxyGV-B | 2.371 (1.441–3.921) | 1.039 (0.520–2.168) | - |
| | PlxyGV-Wn | 5.038 (3.480–7.865) | 2.208 (1.148–6.195) | 2.125 (1.128–5.594) |

genome sequences available in data bases: PlxyGV-C (KU529791.1, 100,980 bp), PlxyGV-K (KU529794.1, 100,978 bp), PlxyGV-T (KU529792.1, 101,004 bp), PlxyGV-M (KU529793.1, 100,986 bp), PlxyGV-SA (KU666537.1, 100,941 bp), and PlxyGV-K1 (NC_002593.1, 100,999 bp). Complete genome sequence alignments show that sequence identities between the genomes of all the PlxyGV isolates, except PlxyGV-SA, are over 99.4% (Fig 2). PlxyGV-SA shares a sequence identity of 98.6% or 98.7% with all the other viral isolates. The sequence identity between the genomes of PlxyGV-C, -K, -M, and -T is as high as 99.9%. PlxyGV-W and PlxyGV-B, similarly, share sequence identities of 99.5% or 99.4% with PlxyGV-C, -K, -M, -T, -K1, and -Wn. PlxyGV-Wn demonstrates an identical sequence identity of 99.6% with PlxyGV-C, -K, -M, -T, and -K1. PlxyGV-K1has a sequence identity of 99.7% with PlxyGV-C, -K, -M, and -T. The frequency of variation in *hr*s is much higher than in other regions. The rates of mutation between PlxyGV-W and the other isolates occurring in *hr*s, other noncoding sequences and ORFs are 53.29 – 96.66, 0.84 – 36.51, and 0.82 – 11.83 per 1,000 bases, respectively (S1 Table). Similar variation frequencies in *hr*s, other noncoding sequences and ORFs are also observed between genomes of the other viral isolates. Base transitions account for most variations between the genomes of the virus isolates. Base transitions are two to three times of transversions between PlxyGV-W and PlxyGV-B, -Wn, -K1, -C, -K, -M, and -T genomes, and nine times of transversions between PlxyGV-W and PlxyGV-SA genomes.

**Table 4. Time-mortality of PlxyGV-W, PlxyGV-B and PlxyGV-Wn for *Plutella xylostella* larve.**

| | Virus | LT$_{50}$±SEM (95%CI)(h) | Log Rank(Mantel-Cox) | | | |
| --- | --- | --- | --- | --- | --- | --- |
| | | | PlxyGV-W | | PlxyGV-B | |
| | | | $\chi2$ | *P* | $\chi2$ | *P* |
| Test 1 | PlxyGV-W | 101±3.758 (93.634–108.366) | - | - | - | - |
| | PlxyGV-B | 113±2.761 (107.589–118.411) | 0.094 | 0.759 | - | - |
| | PlxyGV-Wn | 119±3.779 (111.593–126.407) | 4.093 | 0.043 | 3.598 | 0.058 |
| Test 2 | PlxyGV-W | 117±3.012 (111.097–122.903) | - | - | - | - |
| | PlxyGV-B | 123±2.207 (118.675–127.325) | 3.554 | 0.059 | - | - |
| | PlxyGV-Wn | 123±2.207 (118.675–127.325) | 1.796 | 0.180 | 0.099 | 0.753 |
| Test 3 | PlxyGV-W | 119±1.685 (115.65–122.35) | - | - | - | - |
| | PlxyGV-B | 125±5.271 (117.016–132.984) | 1.014 | 0.314 | - | - |
| | PlxyGV-Wn | 125±6.854 (117.016–132.984) | 2.702 | 0.100 | 1.098 | 0.295 |

**Nucleotides**

| Isolates | -W | -B | -Wn | -C | -K | -T | -M | -SA | -K1 |
|---|---|---|---|---|---|---|---|---|---|
| **PlxyGV-W** |  | 99.8 | 99.5 | 99.4 | 99.5 | 99.5 | 99.5 | 98.6 | 99.4 |
| **PlxyGV-B** | 99.8 |  | 99.5 | 99.5 | 99.5 | 99.5 | 99.5 | 98.6 | 99.5 |
| **PlxyGV-Wn** | 99.6 | 99.7 |  | 99.6 | 99.6 | 99.6 | 99.6 | 98.6 | 99.6 |
| **PlxyGV-C** | 99.6 | 99.7 | 99.8 |  | 99.9 | 99.9 | 99.9 | 98.6 | 99.7 |
| **PlxyGV-K** | 99.7 | 99.7 | 99.8 | 100 |  | 99.9 | 99.9 | 98.7 | 99.7 |
| **PlxyGV-T** | 99.2 | 99.3 | 99.4 | 99.5 | 99.5 |  | 99.9 | 98.7 | 99.7 |
| **PlxyGV-M** | 99.6 | 99.7 | 99.8 | 99.9 | 99.9 | 99.5 |  | 98.7 | 99.7 |
| **PlxyGV-SA** | 99.0 | 99.1 | 99.1 | 99.1 | 99.1 | 98.7 | 99.1 |  | 98.6 |
| **PlxyGV-K1** | 99.4 | 99.5 | 99.5 | 99.5 | 99.5 | 99.1 | 99.5 | 98.9 |  |

(Left vertical axis label: **Amino acids**)

**Fig 2. Identity of genome sequences between nine PlxyGV isolates.** The complete nucleotide sequences of nine PlxyGV genomes and concatenated amino acid sequences encoded by the PlxyGV genomes were aligned separately, and the identity levels between the genomes are expressed as percentage. The amino acid sequences of ORF9, 13, 26, 39, 49, 95 and 108 were not included. W, PlxyGV-W; B, PlxyGV-B; C, PlxyGV-C; K, PlxyGV-K; T, PlxyGV-T; M, PlxyGV-M; SA, PlxyGV-SA; K1, PlxyGV-K1.

Except PlxyGV-K1, all the eight additional PlxyGV isolates have 118 putative protein-coding ORFs in common. ORF organization are completely collinear between the genomes of them. PlxyGV-K1 genome was reported having 120 putative protein-coding ORFs. Sequence alignment shows that the ORF38 and ORF39 of PlxyGV-K1 match the upstream and downstream sequences of the ORF38 in the other isolates, respectively. The difference results from a frameshift induced by a single nucleotide deletion in the genome of PlxyGV-K1 relative to the other PlxyGV isolates. Similarly, the sequence of PlxyGV-K1 ORF48 and ORF49 match the downstream and upstream of the ORF49 (*p74*) of the other PlxyGV isolates. This is also from a single nucleotide insertion/deletion between PlxyGV-K1 and the other isolates. Frameshift variations by single nucleotide changes between PlxyGV-K1 and the other isolates are also found in ORF9, ORF13 (*odv-e18*), ORF26 (*ac23*), ORF95 (*lef3*), and ORF108 resulting in changes in ORF size and predicted amino acid sequences encoded. The ORF9 in PlxyGV-K1 and PlxyGV-SA has extra thirteen codons at the N-terminal relative to its homologs in the other isolates, resulting from a C/T substitution at nt38 upstream of the first ATG of the ORF9 in the other isolates, which creates an new start codon. A single nucleotide missing in the middle of PlxyGV-K1 ORF13 relative to its homologs in the other isolates results in a frameshift after aa48. An A/T substitution in ORF26 creates a stop codon immediate upstream of the second ATG relative to the other PlxyGV isolates. That causes nine codons at the N-terminal missing in PlxyGV-K1 ORF26. A C/A substitution converts the cysteine codon at aa298 in the ORF95 of the other PlxyGV isolates into a stop codon in PlxyGVK1 ORF95, that causes PlxyGV-K1 ORF95 forty aa shorter than its homologs in the other isolates. In addition, PlxyGV-K1 has a cluster of seventeen nucleotides missing in ORF108 relative to the other isolates, after aa137. That causes frameshift and creates a stop codon immediate downstream of the deletion. Whether these differences between PlxyGV-K1 and the other virus isolates result from evolution or sequencing error needs further verification. In addition, there is a cluster of eleven codons inserted within the C-terminal region of PlxyGV-SA ORF50 (*p10*) relative to the ones in the other PlxyGV isolates.

Apart from the ORFs described above, ORF73 and ORF20 are most variable among the ORFs of PlxyGV isolates (Fig 3). ORF73 is homologous to AC91. Homologs of this gene are found in genomes of all Group I alphabaculoviruses and CpGV in addition to PlxyGV [42]. It is rich in proline and serine/threonine residues. The amino acid sequence from AA69 to AA74 of PlxyGV-SA ORF73 is different from the ones of the other isolates while PlxyGV-B ORF73 misses four amino acids in this region. There is a glutamine residue at AA96 (AA92 for PlxyGV-B) position in ORF73 homologs of PlxyGV-W, -B, -Wn, -T, and -K1. Following the AA96 is a long

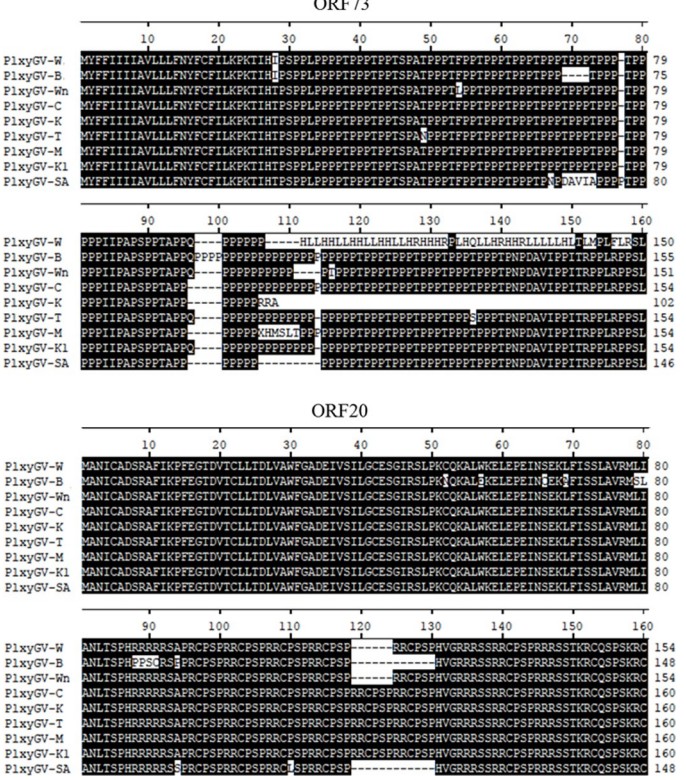

**Fig 3. Sequence alignment of ORF73 and ORF20 homologs of nine PlxyGV isolates.**

cluster of repetitive proline residues varying in number among the virus isolates. PlxyGV-K ORF73 miss all the C-terminal sequences after AA102. The C-terminal of PlxyGV-W ORF73 is totally different from the ones of the other isolates due to frame shift as mentioned before. The ORF20 homologs consist of 223–235 amino acids. There are 4–6 repeated "RCPSPR" and 4 "RC/SP/Q/S/ESPR/H" repeats in the middle region. PlxyGV-B and PlxyGV-SA ORF20 have two copies, PlxyGV-W and PlxyGV-Wn have one copy of "RCPSPR" less than the other isolates.

The identity levels of total amino acid sequences of the prospective protein products, except ORF9, 13, 26, 73, 95 and 108, between the viral genomes are similar to the identity levels of nucleotide sequences (Fig 2). The prospective amino acid sequences of ORF1, 3, 4 (granulin), 12 (AC145), 13 (ODV-E18), 17 (AC29), 27, 29 (PIF3), 33, 41 (PIF7), 53 (P24), 81, 91 (AC76), 100 (FP), 102, 110, 114 (Ubi-lig), and 119 are identical in all of the nine viral isolates (Fig 1). The sizes of individual ORFs of nine PlxyGV isolates and the non-synonymous substitutions per base pair (NSSP) of each ORF are listed in Table 5. As shown in Fig 4, the NSSP levels of most ORFs are between 0 and 5 per thousand; thirty six ORFs have NSSP levels over 5 per thousand, most are functional unknown. ORF19 demonstrates the second highest NSSP (26.67 per thousand). ORF61-ORF72 and ORF74-ORF102 regions seem more conservative than other parts. These regions contain seventeen baculovirus core genes [42] and only one PlxyGV specific gene. Organization in these regions are similar among all baculoviruses [43].

## The genes unique to PlxyGV

There are thirteen ORFs unique to PlxyGV, including ORF1, 3, 5, 19, 22, 27, 33, 58, 81, 105, 108, 111 and 119. These ORFs do not have any additional homologs in data bases. ORF1, 3,

**Table 5. ORF size and number of nonsynonymous mutations in nine PlxyGV isolates.**

| ORF | Name | W | B | Wn | C | T | K | M | SA | K1 | Number of mutations | NSSP (x10⁻³)@ |
|---|---|---|---|---|---|---|---|---|---|---|---|---|
| 1 | | 66 | 66 | 66 | 66 | 66 | 66 | 66 | 66 | 66 | 0 | 0 |
| 2 | P10 | 83 | 83 | 83 | 83 | 83 | 83 | 83 | 83 | 83 | 2 | 8.03 |
| 3 | | 105 | 105 | 105 | 105 | 105 | 105 | 105 | 105 | 105 | 0 | 0 |
| 4 | | 248 | 248 | 248 | 248 | 248 | 248 | 248 | 248 | 248 | 0 | 0 |
| 5 | | 131 | 131 | 131 | 131 | 131 | 131 | 131 | 131 | 131 | 1 | 2.54 |
| 6 | PK | 274 | 274 | 274 | 274 | 274 | 274 | 274 | 274 | 274 | 1 | 1.22 |
| 7 | PIF1 | 536 | 536 | 536 | 536 | 536 | 536 | 536 | 536 | 536 | 5 | 3.11 |
| 8 | | 175 | 175 | 175 | 175 | 175 | 175 | 175 | 175 | 175 | 5 | 9.52 |
| 9 | | **80** | **80** | **80** | **80** | **80** | **80** | **80** | 80 | **93** | 6 | 25 |
| 10 | IE1 | 393 | 393 | 393 | 393 | 393 | 393 | 393 | 393 | 393 | 4 | 3,39 |
| 11 | AC146 | 180 | 180 | 180 | 180 | 180 | 180 | 180 | 180 | 180 | 1 | 1.85 |
| 12 | AC145 | 98 | 98 | 98 | 98 | 98 | 98 | 98 | 98 | 98 | 0 | 0 |
| 13 | ODV-E18 | **80** | **80** | **80** | **80** | **80** | **80** | **80** | 80 | **91** | 0 | 0 |
| 14 | 49K | 446 | 446 | 446 | 446 | 446 | 446 | 446 | 446 | 446 | 3 | 2.24 |
| 15 | EXON0 | 216 | 216 | 216 | 216 | 216 | 216 | 216 | 216 | 216 | 3 | 4.63 |
| 16 | PIF5 | 351 | 351 | 351 | 351 | 351 | 351 | 351 | 351 | 351 | 4 | 3.8 |
| 17 | AC29 | 58 | 58 | 58 | 58 | 58 | 58 | 58 | 58 | 58 | 0 | 0 |
| 18 | | 313 | 313 | 313 | 313 | 313 | 313 | 313 | 313 | 313 | 5 | 5.32 |
| 19 | | 100 | 100 | 100 | 100 | 100 | 100 | 100 | 100 | 100 | 8 | 26.67 |
| 20 | | 229 | 223 | 229 | 235 | 235 | 235 | 235 | 235 | 235 | 14 | 20.93 |
| 21 | P10 | 320 | 320 | 320 | 320 | 320 | 320 | 320 | 320 | 320 | 2 | 2.08 |
| 22 | | **305** | **305** | **305** | **305** | **305** | **305** | **305** | **305** | **305** | 23 | 8.74 |
| 23 | | 131 | 131 | 131 | 131 | 131 | 131 | 131 | 131 | 131 | 1 | 2.54 |
| 24 | | 338 | 338 | 338 | 338 | 338 | 338 | 338 | 338 | 338 | 1 | 0.99 |
| 25 | | **431** | **431** | **431** | **431** | **431** | **431** | **431** | **434** | **431** | 6 | 4.64 |
| 26 | F | **553** | **553** | **553** | **553** | **553** | **553** | **553** | **553** | **544** | 2 | 1.21 |
| 27 | | 79 | 79 | 79 | 79 | 79 | 79 | 79 | 79 | 79 | 0 | 0 |
| 28 | | 210 | 210 | 210 | 210 | 210 | 210 | 210 | 210 | 210 | 2 | 3.17 |
| 29 | PIF3 | 181 | 181 | 181 | 181 | 181 | 181 | 181 | 181 | 181 | 0 | 0 |
| 30 | ODV-E66 | 682 | 682 | 682 | 682 | 682 | 682 | 682 | 682 | 682 | 7 | 3.42 |
| 31 | | 129 | 129 | 129 | 129 | 129 | 129 | 129 | 129 | 129 | 1 | 2.58 |
| 32 | LEF2 | 270 | 270 | 270 | 270 | 270 | 270 | 270 | 270 | 270 | 9 | 12.35 |
| 33 | | 97 | 97 | 97 | 97 | 97 | 97 | 97 | 97 | 97 | 0 | 0 |
| 34 | | 133 | 133 | 133 | 133 | 133 | 133 | 133 | 133 | 133 | 1 | 2.51 |
| 35 | MMP | **402** | **402** | **402** | **402** | **402** | **402** | **402** | **403** | **402** | 4 | 3.32 |
| 36 | P13 | 263 | 263 | 263 | 263 | 263 | 263 | 263 | 263 | 263 | 1 | 1.27 |
| 37 | PIF2 | 368 | 368 | 368 | 368 | 368 | 368 | 368 | 368 | 368 | 9 | 8.15 |
| 38 | | **528** | **528** | **528** | **528** | **528** | **528** | **528** | **528** | **153** | 8 | 5.05 |
| | | | | | | | | | | | **376** | | |
| 40 | AC106/107 | 206 | 206 | 206 | 206 | 206 | 206 | 206 | 206 | 206 | 1 | 1.62 |
| 41 | PIF7 | 53 | 53 | 53 | 53 | 53 | 53 | 53 | 53 | 53 | 0 | 0 |
| 42 | V-UBI | 114 | 114 | 114 | 114 | 114 | 114 | 114 | 114 | 114 | 1 | 2.92 |
| 43 | | 414 | 414 | 414 | 414 | 414 | 414 | 414 | 414 | 414 | 5 | 4.03 |
| 44 | | 130 | 130 | 130 | 130 | 130 | 130 | 130 | 130 | 130 | 6 | 15.38 |
| 45 | 39K | **252** | **252** | **252** | **252** | **252** | **252** | **252** | **251** | **252** | 5 | 6.61 |
| 46 | LEF11 | 96 | 96 | 96 | 96 | 96 | 96 | 96 | 96 | 96 | 1 | 3.47 |
| 47 | SOD | 153 | 153 | 153 | 153 | 153 | 153 | 153 | 153 | 153 | 3 | 6.54 |

*(Continued)*

**Table 5.** (Continued)

| ORF | Name | W | B | Wn | C | T | K | M | SA | K1 | Number of mutations | NSSP (x10$^{-3}$)$^@$ |
|---|---|---|---|---|---|---|---|---|---|---|---|---|
| 49 | PIF0 | **627** | **627** | **627** | **627** | **627** | **627** | **627** | **627** | **51** | 7 | 3.72 |
| | | | | | | | | | | **578** | | |
| 50 | P10 | **135** | **135** | **135** | **135** | **135** | **135** | **135** | **146** | **135** | 6 | 14.81 |
| 51 | P47 | 386 | 386 | 386 | 386 | 386 | 386 | 386 | 386 | 386 | 4 | 3.45 |
| 52 | AC38 | 207 | 207 | 207 | 207 | 207 | 207 | 207 | 207 | 207 | 1 | 1.61 |
| 53 | P24 | 159 | 159 | 159 | 159 | 159 | 159 | 159 | 159 | 159 | 0 | 0 |
| 54 | AC13 | 150 | 150 | 150 | 150 | 150 | 150 | 150 | 150 | 150 | 1 | 2.22 |
| 55 | LEF1 | 251 | 251 | 251 | 251 | 251 | 251 | 251 | 251 | 251 | 1 | 1.33 |
| 56 | FGF | 221 | 221 | 221 | 221 | 221 | 221 | 221 | 221 | 221 | 2 | 3.02 |
| 57 | | 100 | 100 | 100 | 100 | 100 | 100 | 100 | 100 | 100 | 4 | 13.33 |
| 58 | | 53 | 53 | 53 | 53 | 53 | 53 | 53 | 53 | 53 | 1 | 6.29 |
| 59 | AC150 | 79 | 79 | 79 | 79 | 79 | 79 | 79 | 79 | 79 | 2 | 8.44 |
| 60 | LEF6 | 86 | 86 | 86 | 86 | 86 | 86 | 86 | 86 | 86 | 2 | 7.75 |
| 61 | DBP | 263 | 263 | 263 | 263 | 263 | 263 | 263 | 263 | 263 | 2 | 2.53 |
| 62 | | **136** | **136** | **136** | **136** | **136** | **136** | **136** | **138** | **136** | 1 | 2.45 |
| 63 | AC103 | 377 | 377 | 377 | 377 | 377 | 377 | 377 | 377 | 377 | 2 | 1.77 |
| 64 | AC102 | 96 | 96 | 96 | 96 | 96 | 96 | 96 | 96 | 96 | 2 | 6.94 |
| 65 | | 152 | 152 | 152 | 152 | 152 | 152 | 152 | 152 | 152 | 3 | 6.58 |
| 66 | AC101 | 366 | 366 | 366 | 366 | 366 | 366 | 366 | 366 | 366 | 3 | 2.73 |
| 67 | P6.9 | 56 | 56 | 56 | 56 | 56 | 56 | 56 | 21T | 56 | 1 | 5.95 |
| 68 | AC145/150 | **66** | **66** | **66** | **66** | **66** | **66** | **66** | **65** | **66** | 1 | 5.05 |
| 69 | LEF5 | 247 | 247 | 247 | 247 | 247 | 247 | 247 | 247 | 247 | 2 | 2.70 |
| 70 | 38K | 340 | 340 | 340 | 340 | 340 | 340 | 340 | 340 | 340 | 4 | 3.92 |
| 71 | PIF4 | 161 | 161 | 161 | 161 | 161 | 161 | 161 | 161 | 161 | 2 | 4.14 |
| 72 | HEL1 | 1124 | 1124 | 1124 | 1124 | 1124 | 1124 | 1124 | 1124 | 1124 | 9 | 2.67 |
| 73 | AC91 | **153** | **159** | **155** | **158** | **102** | **158** | **150** | **158** | **158** | 75 | 158 |
| 74 | ODV-E25 | 214 | 214 | 214 | 214 | 214 | 214 | 214 | 214 | 214 | 1 | 1.56 |
| 75 | AC93 | 156 | 156 | 156 | 156 | 156 | 156 | 156 | 156 | 156 | 2 | 4.27 |
| 76 | AC92 | 250 | 250 | 250 | 250 | 250 | 250 | 250 | 250 | 250 | 1 | 1.33 |
| 77 | | 135 | 135 | 135 | 135 | 135 | 135 | 135 | 135 | 135 | 2 | 4.94 |
| 78 | LEF4 | 432 | 432 | 432 | 432 | 432 | 432 | 432 | 432 | 432 | 4 | 3.09 |
| 79 | VP39 | 320 | 320 | 320 | 320 | 320 | 320 | 320 | 320 | 320 | 1 | 1.04 |
| 80 | ODV-EC27 | 287 | 287 | 287 | 287 | 287 | 287 | 287 | 287 | 287 | 5 | 5.81 |
| 81 | | 107 | 107 | 107 | 107 | 107 | 107 | 107 | 107 | 107 | 0 | 0 |
| 82 | | 340 | 340 | 340 | 340 | 340 | 340 | 340 | 340 | 340 | 7 | 6.86 |
| 83 | | 69 | 69 | 69 | 69 | 69 | 69 | 69 | 69 | 69 | 3 | 14.49 |
| 84 | Pif8 | 533 | 533 | 533 | 533 | 533 | 533 | 533 | 533 | 533 | 7 | 4.38 |
| 85 | TLP20 | 139 | 139 | 139 | 139 | 139 | 139 | 139 | 139 | 139 | 2 | 4.80 |
| 86 | AC81 | 191 | 191 | 191 | 191 | 191 | 191 | 191 | 191 | 191 | 1 | 1.75 |
| 87 | GP41 | 283 | 283 | 283 | 283 | 283 | 283 | 283 | 283 | 283 | 4 | 4.71 |
| 88 | AC78 | 89 | 89 | 89 | 89 | 89 | 89 | 89 | 89 | 89 | 1 | 3.75 |
| 89 | VLF1 | 346 | 346 | 346 | 346 | 346 | 346 | 346 | 346 | 346 | 4 | 3.85 |
| 90 | | 175 | 175 | 175 | 175 | 175 | 175 | 175 | 175 | 175 | 4 | 7.62 |
| 91 | AC76 | 81 | 81 | 81 | 81 | 81 | 81 | 81 | 81 | 81 | 0 | 0 |
| 92 | AC75 | 145 | 145 | 145 | 145 | 145 | 145 | 145 | 145 | 145 | 1 | 2.30 |
| 93 | DNAPOL | **978** | **978** | **978** | **978** | **978** | **978** | **978** | **978** | **979** | 10 | 3.41 |
| 94 | AC66 | 651 | 651 | 651 | 651 | 651 | 651 | 651 | 651 | 651 | 6 | 3.07 |

(*Continued*)

**Table 5.** (Continued)

| ORF | Name | W | B | Wn | C | T | K | M | SA | K1 | Number of mutations | NSSP (x10⁻³)@ |
|-----|------|---|---|----|---|---|---|---|----|----|---------------------|---------------|
| 95 | LEF3 | **337** | **337** | **337** | **337** | **337** | **337** | **337** | **337** | **297** | 4 | 3.96 |
| 96 | PIF6 | 128 | 128 | 128 | 128 | 128 | 128 | 128 | 128 | 128 | 1 | 2.60 |
| 97 | | 194 | 194 | 194 | 194 | 194 | 194 | 194 | 194 | 194 | 3 | 5.15 |
| 98 | IAP | 281 | 281 | 281 | 281 | 281 | 281 | 281 | 281 | 281 | 5 | 5.93 |
| 99 | LEF9 | 494 | 494 | 494 | 494 | 494 | 494 | 494 | 494 | 494 | 4 | 2.70 |
| 100 | FP | 138 | 138 | 138 | 138 | 138 | 138 | 138 | 138 | 138 | 0 | 0 |
| 101 | DNA-LIG | 523 | 523 | 523 | 523 | 523 | 523 | 523 | 523 | 523 | 3 | 1.91 |
| 102 | | 61 | 61 | 61 | 61 | 61 | 61 | 61 | 61 | 61 | 0 | 0 |
| 103 | | **63** | **63** | **66** | **66** | **66** | **66** | **66** | **66** | **66** | 2 | 10.58 |
| 104 | FGF | 396 | 396 | 396 | 396 | 396 | 396 | 396 | 396 | 396 | 4 | 3.37 |
| 105 | | 214 | 214 | 214 | 214 | 214 | 214 | 214 | 214 | 214 | 5 | 6.23 |
| 106 | ALK-EXO | 378 | 378 | 378 | 378 | 378 | 378 | 378 | 378 | 378 | 4 | 3.53 |
| 107 | HEL2 | 436 | 436 | 436 | 436 | 436 | 436 | 436 | 436 | 436 | 1 | 0.76 |
| 108 | | **281** | **281** | **281** | **281** | **281** | **281** | **281** | **281** | **138** | 3 | 3.56 |
| 109 | LEF8 | 838 | 838 | 838 | 838 | 838 | 838 | 838 | 838 | 838 | 6 | 2.39 |
| 110 | | 114 | 114 | 114 | 114 | 114 | 114 | 114 | 114 | 114 | 0 | 0 |
| 111 | | 192 | 192 | 192 | 192 | 192 | 192 | 192 | 192 | 192 | 3 | 5.21 |
| 112 | UBI-LIG | 137 | 137 | 137 | 137 | 137 | 137 | 137 | 137 | 137 | 4 | 9.73 |
| 113 | | **198** | **198** | **198** | **197** | **198** | **197** | **197** | **197** | **198** | 9 | 15.15 |
| 114 | | 109 | 109 | 109 | 109 | 109 | 109 | 109 | 109 | 109 | 0 | 0 |
| 115 | VP1054 | 311 | 311 | 311 | 311 | 311 | 311 | 311 | 311 | 311 | 3 | 3.22 |
| 116 | | 59 | 59 | 59 | 59 | 59 | 59 | 59 | 59 | 59 | 2 | 11.30 |
| 117 | | 249 | 249 | 249 | 249 | 249 | 249 | 249 | 249 | 249 | 2 | 2.68 |
| 118 | EGT | 429 | 429 | 429 | 429 | 429 | 429 | 429 | 429 | 429 | 6 | 4.66 |
| 119 | | 142 | 142 | 142 | 142 | 142 | 142 | 142 | 142 | 142 | 0 | 0 |
| 120 | ME53 | 308 | 308 | 308 | 308 | 308 | 308 | 308 | 308 | 308 | 1 | 1.08 |

@ Variations in PlxyGV-K1 ORF13, 26, 38, 39, 48, 49, 95 and 108 were not count in.

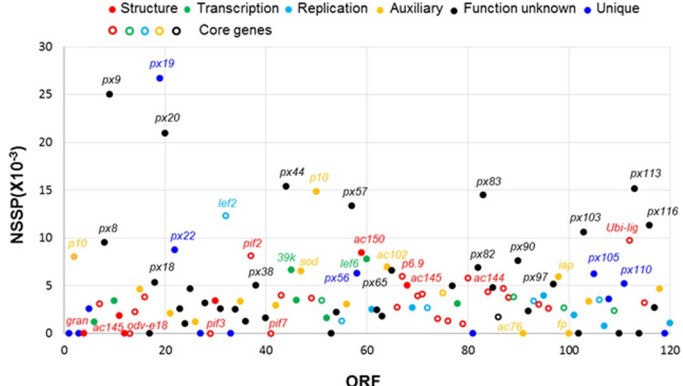

**Fig 4. Nonsynonymous variations in ORFs of nine PlxyGV isolates.** Scatter plots represent ORFs with their levels of nonsynonymous sequence substitution per base pair (NSS/bp ×10⁻³) demonstrated. Red, green, light blue, orange, black and blue colors indicate structure protein, transcription factor, DNA replication factor, auxiliary, function unknown and PlxyGV unique genes, respectively. Hollow circles indicate baculovirus core genes. ORF71, which shows 158 NSS per base pair (×10⁻³) is not included.

27, 33, 81, and 119 have identical predicted amino acid sequences among the nine viral isolates. The transcription start sites (TSS) and the polyadenylation sites (PAS) of these PlxyGV-specific genes were determined by Rapid Amplification of cDNA Ends analysis, using total RNAs purified from *P. xylostella* larvae infected by PlxyGV-Wn. As shown in Fig 5, ORF27 and ORF19 have three and two TSSs separately. The other ORFs have a single TSS. Notably, the TSS of ORF22 is located at 191 nt downstream of the first ATG and 80 nt upstream of the third ATG. The sequences between the two ATGs are highly variable among ORF22 homologs of the PlxyGV isolates. This suggests that the third ATG is the real start codon of ORF22. Similarly, the TSS of ORF105 were localized to 306 nt downstream of the first ATG and 254 nt upstream of the second ATG, implying that the second ATG is likely the real start codon. A TATA box, and A(A/T)CGT(G/T) and CGTGC motifs are present in regions between 40 nt upstream of TSS and initiation ATG, and a baculovirus late promoter motif A/G/TTAAG is located within 100 nt upstream of initiation ATG of individual ORFs are shown in Fig 5 (and S2 Table). Predicted RNA polymerase II TSS motifs CAGT/CAAT/CATT locating between TSS and the initiation ATG are also listed in S2 Table. None of the TSS identified were associated with the TAAG motifs although there are some present in ORF3 (-13), ORF 5 (-73), ORF19 (-13 and ORF58 (-91, and -25). As shown in Fig 5, the TSS of ORF33 and ORF111 extend to upstream of ORF34 and ORF110 respectively.

All the PlxyGV-specific ORFs have single PAS (Fig 5 and S2 Table). The PAS of ORF3 and ORF58 are located downstream of ORF4 and ORF57 respectively. In most cases, there is one or two transcription termination signal elements AATAAA or ATTAAA (ORF27) near the PAS. ORF108 lacks a typical AATAAA or ATTAAA. However there is an AATTAAT located 23 nt upstream of the PAS.

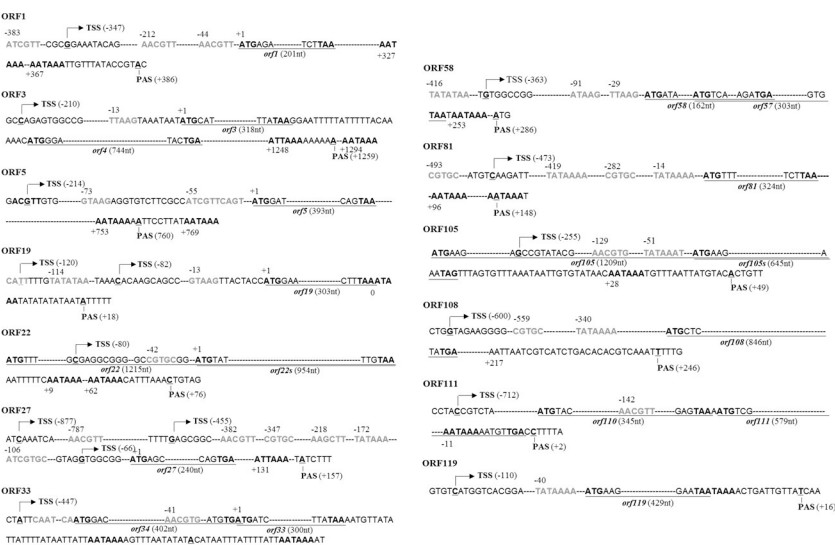

**Fig 5. Structures of the transcription cassettes of PlxyGV-specific genes.** The transcription start sites are in bold and underlined, and transcription direction are indicated by arrows. The major promotor elements, TATA boxes, A(A/T) CGT(G/T), CGTGC, and A/G/TTAAG motifs are present in grey bold. Their positions relative to the initiation ATG of the individual PlxyGV-specific ORFs are shown by the numbers above. The initiation ATG and stop codons are in black bold and underlined. The transcription terminator elements, AATAAA and ATTAAA are in black bold and their positions relative to the stop codons of individual PlxyGV-specific ORFs are indicated by the number underneath.

## Phylogenetic tree of PlxyGV isolates

An evolutionary tree of the nine virus isolates was constructed using MEGA6 software with the neighbor joining method, based on concatenated amino acid sequences encoded by the thirty eight baculovirus core genes [42], using Hyphantria cunea granulovirus (HycuGV) as an outgroup. HycuGV was shown to be most close to PlxyGV [44]. In the process, PlxyGV-K1 ORF48 and ORF49 were merged into one ORF by filling the single missing base relative to their homologs in the other viral isolates. It can be seen that PlxyGV-C clusters with PlxyGV-K. This cluster is near PlxyGV-M and PlxyGV-T. PlxyGV-W and PlxyGV-B are located in the same cluster. PlxyGV-Wn is more distant from PlxyGV-W and PlxyGV-B than the other isolates except PlxyGV-SA. PlxyGV-K1 is closer to PlxyGV-T, -M, -C and -K than PlxyGV-Wn although they are in the same clade. PlxyGV-SA is relatively distant from the other isolates.

## Discussion

In this study, we describe the genome sequencing and analysis of three PlxyGV isolates. PlxyGV-W was first isolated from a diamondback moth larva from cabbage fields in Wuhan city, in early 1980s. PlxyGV-B was originally isolated near Beijing in the early 1980s. The PlxyGV-Wn was isolated in Wuhan, in April, 2018. The sequence data show that the genomes of PlxyGV-W and PlxyGV-B share a sequence identity of 99.8%. And the amino acid sequences encoded by these two viral genomes are almost identical except for variations in ORF20 and ORF73. Surprisingly, although PlxyGV-Wn was isolated from the same area as PlxyGV-W, it shares higher sequence identity with PlxyGV-K1, -T, -C, -K and -M than with PlxyGV-W and -B (Fig 2), and is more closely related to PlxyGV-K1, -T, -C, -K and -M than to PlxyGV-W and -B as demonstrated by the phylogenetic tree (Fig 6). It implies that PlxyGV-Wn and PlxyGV-W originated from different populations, which may have emerged in Wuhan at different time points in history. Located at the junction of the Yantze and Han rivers, Wuhan is a transportation hub that facilitates the introduction of species from different regions. In addition, PlxyGV-W and -B demonstrated higher insecticide activity to diamondback moth larvae than PlxyGV-Wn. How the genomic sequence variations determine the differential insecticidal activity between PlxyGV-Wn and the other two virus isolates will require further investigation. Notably, among the fouty eight ORFs containing non-synonymous variations between PlxyGV-Wn and PlxyGV-W and/or PlxyGV-B are homologs of *egt* and six *per os* infectivity factor genes *pif0*, *pif1*, *pif2*, *pif5*, *pif6 and pif8* and *odv-e66*, an additional possible *per os* infectivity factor gene. *Egt* encodes ecdysteroid UDP-glucosyltransferase to block molting and pupation in infected larvae, thereby to prolong the feeding stage of infected larvae [45, 46]. *per os* infectivity factors are required for infection of insects [47–50].

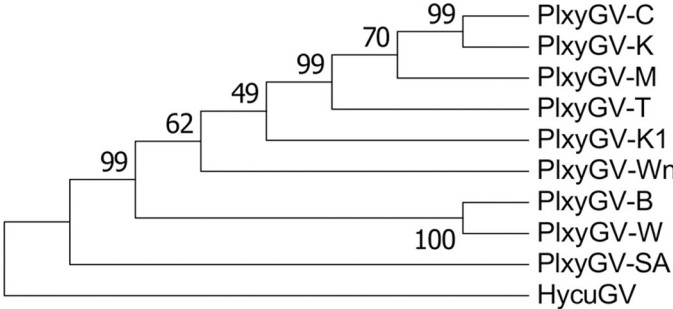

**Fig 6. Neighbor joining phylogenetic tree of PlxyGV isolates.** The phylogeny was inferred using concatenated amino acid sequences of the homologs of thirty eight baculovirus core genes of nine PlxyGV isolates.

Previously, intra isolate genetic diversity was reported in many baculoviruses. Phenotypic changes were observed between genotypes. For instance, twenty-five genotypic variants of a nucleopolyhedrovirus were identified and purified from a single *Panolis flammea* larva. Four of the genotypic variants were found having significant difference in pathogenicity, speed of killing and yield [2]. Genome sequencing makes it possible to characterize inter and intra isolate diversity of same species. Complete NPs contained in a baculovirus genome was first identified in Mamestra configurata nucleopolyhedrovirus v90/4 [4]. It is considered that presence of a pool of polymorphisms may provide advantage in adapting to a changeable environment. In this study, intra isolate NPs are identified in the genomes of all the three PlxyGV isolates. The NPs occurring in PlxyGV-W almost completely overlap with the ones in PlxyGV-B, although PlxyGV-B has more nucleotide polymorphisms than PlxyGV-W. Notably, the ORFs containing non-synonymous polymorphisms include homologs of three DNA replication factors HEL1, LEF5 and DBP, two late expression factors LEF8 and LEF9, and *per os* infectivity factor PIF9. The NP profile in PlxyGV-Wn genome is totally different from the ones in the other two isolates. The ORFs with non-synonymous polymorphisms in PlxyGV-Wn genome also include a homolog of *per os* infectivity factor, PIF5.

Genome sequence comparison of nine PlxyGV isolates reveals high genetic stability of PlxyGV. These PlxyGV isolates are from five areas of four countries, but they have limited variations in genome size and nucleotide sequence. The maximal length difference is only sixty three base pairs, which exists between PlxyGV-W/PlxyGV-SA (100,941 bp) and PlxyGV-T (101,004 bp). The minimum sequence homogeneity is 98.6 percent, existing between PlxyGV-SA and four isolates from the mainland of China and the one from Japan. No gain/loss of prospective protein-coding ORFs identified among the viral isolates. The high genetic stability of PlxyGV ensures the stability and specificity of its control effect on diamondback moth, and is helpful to commercialization of PlxyGV insecticides. In addition, it also facilitates the construction of recombinant PlxyGVs with enhanced insecticidal activity through genetic manipulation, ensuring that the superior properties obtained by engineered viruses are not easily lost or changed.

Previously reported genomes of different geographic isolates of the same baculovirus species usually have variations in gene contents, frequently occurring in *bro* gene associated regions [4, 51]. PlxyGV lacks *bro* homologs. Similarly, seven Erinnyis ello granulovirus (ErelGV) field isolates also have common ORF contents and organization, but all of them are isolated in Brazil [7]. Similar to other baculoviruses, NPs are mainly present in *hr*s and two ORFs containing repetitive sequences in PlxyGV genome. All the ORFs with the highest levels of nonsynonymous mutations have unknown functions. Paralogous genes *p10* and *ac145/ac150* homologs demonstrate relatively high levels of non-synonymous mutations. Baculoviruses have thirty eight core genes whose homologs are present in all baculovirus genomes sequenced to date [42]. Generally, the PlxyGV core gene homologs contain low levels of non-synonymous variations among the nine viral isolates. Similar phenomenon were also observed in ErelGV isolates [7].

Thirteen ORFs specific to PlxyGV are conserved in all the PlxyGV isolates. The TSS and PAS of the ORFs were identified by RACE analysis. The data suggest that all these PlxyGV-specific ORFs are transcribed during infection. Seven of these PlxyGV unique ORFs have no non-synonymous variation among all the PlxyGV isolates, implying these genes must play important roles in replication and infection of PlxyGV. Notably, none of the PlxyGV-specific genes were found to start transcription from late promotor motifs. We are not sure whether these results reflected the real situation. If the levels of some transcripts starting from TAAG motifs were very low, they might not be detected.

The PlxyGV isolates analyzed in this study are from five geographically separate areas, the mainland of China, Taiwan, Japan, Malaysia and South Africa. Phylogenetic analysis shows that PlxyGV-SA is distantly related to the other isolates, which may reflect their geographic

distance from the other isolates. PlxyGV-M from Malaysia, PlxyGV-C from mainland China and the two isolates from Taiwan, PlxyGV-K and -T are closely related. However, PlxyGV-C is distantly related to other three isolates from the mainland of China. It is likely some isolates migrated from one area to another area recently.

## Supporting information

**S1 Table. Mutation frequency of eight PlxyGV isolates relative to the PlxyGV-W genome sequence in coding, noncoding and *hr* regions ($\times 10^{-3}$).**
(DOCX)

**S2 Table. The transcription start sites and polyadenylation sites of PlxyGV specific genes.**
(DOCX)

## Acknowledgments

We thank Professor Qin, Qi-Lian and Professor Zhang, Huan for their kind support and technical assistance in GV counting. Thank professor George F Rohrmann for critical reviewing the manuscript.

## Author Contributions

**Conceptualization:** Lu-Lin Li.

**Data curation:** Man-Li Zhang, Lu-Lin Li.

**Formal analysis:** Lu-Lin Li.

**Funding acquisition:** Lu-Lin Li.

**Investigation:** Man-Li Zhang, Ling-Qian Wang, Yong Qi, Dong-Hui Zhou, Lu-Lin Li.

**Project administration:** Lu-Lin Li.

**Resources:** Man-Li Zhang, Yi Wu, Lu-Lin Li.

**Supervision:** Lu-Lin Li.

**Writing – original draft:** Man-Li Zhang, Lu-Lin Li.

**Writing – review & editing:** Lu-Lin Li.

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
