## [Decision Letter · Decision Letter 0]

16 Sep 2020

PONE-D-20-20094

Comparative genomic analysis of three geographical isolates from China reveals high genetic stability of Plutella xylostella granulovirus

PLOS ONE

Dear Dr. Li,

Thank you for submitting your manuscript to PLOS ONE. After careful consideration, we feel that it has merit but does not fully meet PLOS ONE’s publication criteria as it currently stands. Therefore, we invite you to submit a revised version of the manuscript that addresses the points raised during the review process.

We look forward to receiving your revised manuscript.

Kind regards,

Xi Zhou, Ph.D.

Academic Editor

PLOS ONE

Journal Requirements:

Reviewers' comments:

Reviewer's Responses to Questions

**Comments to the Author**

1. Is the manuscript technically sound, and do the data support the conclusions?

Reviewer #1: Yes

Reviewer #2: Yes

2. Has the statistical analysis been performed appropriately and rigorously? 

Reviewer #1: Yes

Reviewer #2: Yes

3. Have the authors made all data underlying the findings in their manuscript fully available?

Reviewer #1: Yes

Reviewer #2: Yes

4. Is the manuscript presented in an intelligible fashion and written in standard English?

Reviewer #1: Yes

Reviewer #2: Yes

5. Review Comments to the Author

Reviewer #1: Zhang et al present a work in which they comparatively analyzed sequences of PlxyGV isolates, providing some valid information about genetic stability and diversity of baculovirus PlxyGV. In addition, they determined the transcription and polyadenylation sites of some PlxyGV ORFs. Specific points were as follows.

1. Line 16. The words "from near" should be corrected using an English statement.

2. Line 19. Change the location of the word "respectively".

3. Line 32-33. Correct to "the transcription and polyadenylation sites of…"

4. Line 49 and 52. Correct "the same".

5. Line 54-58. This is an example to testify "…facilitate the improvement of baculovirus pesticides by mixing different virus genotypes…".

6. Line 59. The abbreviation "PlxyGV" (including others) was written above.

7. Line 60-61. This sentence is confusing and corrected to "…Plutella xylostella, a destructive and widely distributed pest".

8. Line 59-71. This paragraph is redundant and not readable. It is proposed that Plutella xylostella is stated first (destructive, widely distributed, resistance to chemical pesticides, etc.), and then PlxyGV is explained.

9. Line 72-75. The two sentences were corrected to "The complete genome sequence of a PlxyGV isolate (PlxyGV-K1) from Japan was firstly reported in 2000 to consist of 100,999 bp, and encode 120 putative protein-coding ORFs. Subsequently, the genomic sequences of five additional isolates…", to avoid some repeated words.

10. Line 79-80. Remove sentences "PlxyGV-W and PlxyGV-B were isolated in Wuhan and Beijing forty years ago, while PlxyGV-Wn was collected recently in Wuhan". They are stated in section Materials and Methods.

11. Line 91-94. The sentence "PlxyGV-B is from a commercialized biopesticide, that was obtained from the Institute of Zoology, Chinese Academy of Sciences in Beijing and was originally isolated in Beijing in 1980s (Qin, Qi-Lian personal communication)" is confusing.

12. The section 2.1 Virus and insects. Authors explained origin and propagation of PlxyGV, but did not provide some statements about the insects.

13. Line 97-109. Purification of OBs and extraction of DNA (including the following RNA purification and RACE analysis, Bioassays) were common methods in baculovirus field. Authors could cite some references unless there are some modifications.

14. Line 126-127. The sentence is incomplete. In addition, there is a space between 24 and h, etc.

15. In legend of Fig. 1, correct "location" to "locations".

16. Line 235. Correct "correspondent" to "corresponding".

17. Line 250. Change "∨" indicating insertion. Someone may make a mistake to consider it as valine.

18. Line 262, 338, 344, etc. Correct "C-terminal" to "C-terminus".

19. Line 181-282. Correct "and determining LC50 and ST50 in bioassays". Perhaps, they are grammar mistakes.

20. In reviewer’s opinion, the organization of Results could be changed. The results of larvae bioassays could be stated first to make an importance of "facilitate the improvement of baculovirus pesticides by mixing different virus genotypes".

21. Check the References and rewrite them in a uniform format according to PLOS ONE required style.

Reviewer #2: The current study is related to Comparative genomic analysis of three geographical isolates from China reveals high genetic stability of Plutella xylostella granulovirus. They sequenced and comparatively analyzed to investigate genetic stability and diversity of PlxyGV，PlxyGV-W, PlxyGV-B. By alignment of the genome sequences of nine PlxyGV isolates sequenced up to date, the author found that PlxyGV genome is fairly stable in nature.

Some aspects should be addressed to improve this manuscript.

1.PlxyGV-W and PlxyGV-B demonstrate higher insecticidal activity than PlxyGV-Wn for P. xylotella larvae. The LC50 of PlxyGV-Wn is about two times of the ones of the other two virus isolates while there is no significant difference between PlxyGV-W and PlxyGV-B. Could you analyze the reasons for different insecticidal virulence of the three GV isolates from the perspective of insect virus genome?

2.By alignment of the genome sequences of nine PlxyGV isolates sequenced, the author found that PlxyGV genome is fairly stable in nature. What is the enlightenment and significance of your current work for the next step in the development of recombinant viruses or insecticides with higher insecticidal virulence against Plutella xylostella?

6. PLOS authors have the option to publish the peer review history of their article (what does this mean?). If published, this will include your full peer review and any attached files.

Reviewer #1: **Yes: **Zhong-Jian Guo

Reviewer #2: No

---

## [Author Response · Author response to Decision Letter 0]

26 Oct 2020

Responses to reviewer #1

1. Line 16. The words "from near" should be corrected using an English statement.

Answer: It seems to me it is OK.

2. Line 19. Change the location of the word "respectively".

Answer: Its position seems correct.

3. Line 32-33. Correct to "the transcription and polyadenylation sites of…"

Thanks! It has been changed to “the transcription start sites and polyadenylation sites of thirteen PlxyGV-specific ORFs, …”.

4. Line 49 and 52. Correct "the same".

Answer: “the same” in line 49 has been changed to “some”. The one in line seems correct.

5. Line 54-58. This is an example to testify "…facilitate the improvement of baculovirus pesticides by mixing different virus genotypes…".

Answer: Yes, it is an example.

6. Line 59. The abbreviation "PlxyGV" (including others) was written above.

Answer: It is present in text at first time, although it was present in abstract.

7. Line 60-61. This sentence is confusing and corrected to "…Plutella xylostella, a destructive and widely distributed pest".

Answer: It has been changed to “…Plutella xylostella, a major destructive pest of cruciferous crops worldwide”

8. Line 59-71. This paragraph is redundant and not readable. It is proposed that Plutella xylostella is stated first (destructive, widely distributed, resistance to chemical pesticides, etc.), and then PlxyGV is explained.

Answer: The second and third sentences (Line 59-61) have been rewritten as “It is pathogenic for the diamondback moth, Plutella xylostella, a major destructive pest of cruciferous crops worldwide [11]. The virus has been isolated in…”.

9. Line 72-75. The two sentences were corrected to "The complete genome sequence of a PlxyGV isolate (PlxyGV-K1) from Japan was firstly reported in 2000 to consist of 100,999 bp, and encode 120 putative protein-coding ORFs. Subsequently, the genomic sequences of five additional isolates…", to avoid some repeated words.

Answer: Thanks for your suggestion. The sentences have rewritten accordingly. 

10. Line 79-80. Remove sentences "PlxyGV-W and PlxyGV-B were isolated in Wuhan and Beijing forty years ago, while PlxyGV-Wn was collected recently in Wuhan". They are stated in section Materials and Methods.

Answer: The sentence has been removed.

11. Line 91-94. The sentence "PlxyGV-B is from a commercialized biopesticide, that was obtained from the Institute of Zoology, Chinese Academy of Sciences in Beijing and was originally isolated in Beijing in 1980s (Qin, Qi-Lian personal communication)" is confusing.

Answer: The sentence has been changed to “PlxyGV-B is from a commercialized biopesticide, that was originally isolated in Beijing in 1980s (Qin, Qi-Lian personal communication).”

12. The section 2.1 Virus and insects. Authors explained origin and propagation of PlxyGV, but did not provide some statements about the insects.

Answer: It is described as “…third instar laboratory reared Diamondback moth larvae”

13. Line 97-109. Purification of OBs and extraction of DNA (including the following RNA purification and RACE analysis, Bioassays) were common methods in baculovirus field. Authors could cite some references unless there are some modifications.

Answer: For OB purification, a reference cited has been added. For RACE and bioassays, references have already been described. 

14. Line 126-127. The sentence is incomplete. In addition, there is a space between 24 and h, etc.

Answer: The sentence has been rewritten as “P. xylostella larvae in third instar were infected with PlxyGV-Wn by feeding with viral OBs-contaminated diet and collected at 12 h, 24 h, 48 h, 72 h and 96 h post infection.”

15. In legend of Fig. 1, correct "location" to "locations".

Answer: They have been corrected accordingly.

16. Line 235. Correct "correspondent" to "corresponding".

Answer: The sentence has been changed to “The hr1, hr2, hr3, and hr4 of PlxyGV-B are 40 bp, 2 bp, 3 bp and 15 bp longer than those of PlxyGV-W, respectively.”

17. Line 250. Change "∨" indicating insertion. Someone may make a mistake to consider it as valine.

Answer: I am not sure it is necessary to make change. The symbol “∨” looks different from the letter “V” in the table.

18. Line 262, 338, 344, etc. Correct "C-terminal" to "C-terminus".

Answer: These are right, do not need to correct.

19. Line 181-282. Correct "and determining LC50 and ST50 in bioassays". Perhaps, they are grammar mistakes.

Answer: I do not see any mistake in this sentence.

20. In reviewer’s opinion, the organization of Results could be changed. The results of larvae bioassays could be stated first to make an importance of "facilitate the improvement of baculovirus pesticides by mixing different virus genotypes".

Answer: Good ideal. The results of bioassays can be put either first or last. 

21. Check the References and rewrite them in a uniform format according to PLOS ONE required style.

Answer: The references have been reformatted accordingly.

Response to reviewer 2 

1. PlxyGV-W and PlxyGV-B demonstrate higher insecticidal activity than PlxyGV-Wn for P. xylotella larvae. The LC50 of PlxyGV-Wn is about two times of the ones of the other two virus isolates while there is no significant difference between PlxyGV-W and PlxyGV-B. Could you analyze the reasons for different insecticidal virulence of the three GV isolates from the perspective of insect virus genome?

Answer: Thanks for the comments, the sentences below have been added at the end of the first paragraph in “Discussion” section. 

“Notably, among the fouty eight ORFs containing non-synonymous variations between PlxyGV-Wn and PlxyGV-W and/or PlxyGV-B are homologs of egt and six per os infectivity factor genes pif0, pif1, pif2, pif5, pif6 and pif8 and odv-e66, an additional possible per os infectivity factor gene. Egt encodes ecdysteroid UDP-glucosyltransferase to block molting and pupation in infected larvae, thereby to prolong the feeding stage of infected larvae[45,46]. per os infectivity factor are required for infection of insects[47-49]”

2. By alignment of the genome sequences of nine PlxyGV isolates sequenced, the author found that PlxyGV genome is fairly stable in nature. What is the enlightenment and significance of your current work for the next step in the development of recombinant viruses or insecticides with higher insecticidal virulence against Plutella xylostella? 

Answer: The sentences below have been added in the paragraph 3 (line 496-501). “The high genetic stability of PlxyGV ensures the stability and specificity of its control effect on diamondback moth, and is helpful to commercialization of PlxyGV insecticides. In addition, high genetic stability also facilitates the construction of recombinant viruses with enhanced insecticidal activity through genetic manipulation, ensuring that the superior properties obtained by engineered viruses are not easily lost or changed.”

---

## [Editor Report · Decision Letter 1]

17 Nov 2020

Comparative genomic analysis of three geographical isolates from China reveals high genetic stability of Plutella xylostella granulovirus

PONE-D-20-20094R1

Dear Dr. Li,

We’re pleased to inform you that your manuscript has been judged scientifically suitable for publication and will be formally accepted for publication once it meets all outstanding technical requirements.

Kind regards,

Xi Zhou, Ph.D.

Academic Editor

PLOS ONE
---

## [Editor Report · Acceptance letter]

5 Jan 2021

PONE-D-20-20094R1 

Comparative genomic analysis of three geographical isolates from China reveals high genetic stability of Plutella xylostella granulovirus 

Dear Dr. Li:

I'm pleased to inform you that your manuscript has been deemed suitable for publication in PLOS ONE. Congratulations! Your manuscript is now with our production department. 

Kind regards, 

on behalf of

Prof. Xi Zhou 

Academic Editor

PLOS ONE